# Learning to Open and Traverse Doors
# with a Legged Manipulator

**Mike Zhang, Yuntao Ma, Takahiro Miki, and Marco Hutter**
Robotic Systems Lab, ETH Zurich, Switzerland

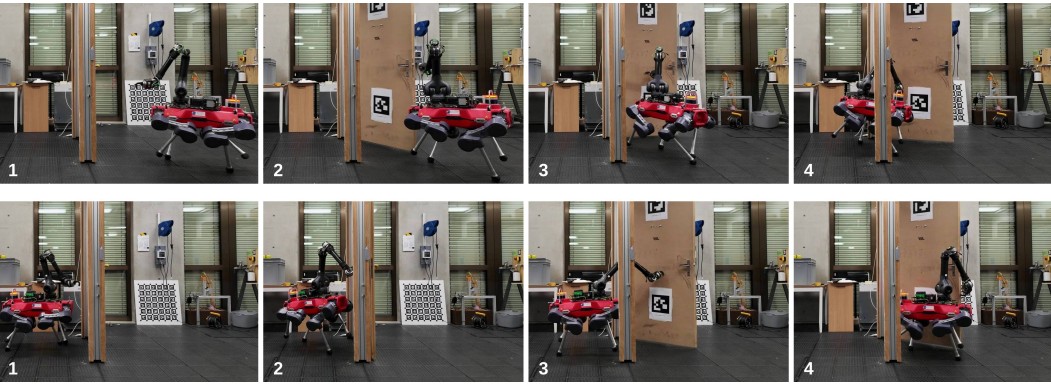

Figure 1: We present a control policy that can open and pass through both pull (top) and push (bottom) doors by estimating the door properties during deployment.

**Abstract:** Using doors is a longstanding challenge in robotics and is of significant practical interest in giving robots greater access to human-centric spaces. The task is challenging due to the need for online adaptation to varying door properties and precise control in manipulating the door panel and navigating through the confined doorway. To address this, we propose a learning-based controller for a legged manipulator to open and traverse through doors. The controller is trained using a teacher-student approach in simulation to learn robust task behaviors as well as estimate crucial door properties during the interaction. Unlike previous works, our approach is a single control policy that can handle both push and pull doors through learned behaviour which infers the opening direction during deployment without prior knowledge. The policy was deployed on the ANYmal legged robot with an arm and achieved a success rate of 95.0% in repeated trials conducted in an experimental setting. Additional experiments validate the policy's effectiveness and robustness to various doors and disturbances. A video overview of the method and experiments can be found at `youtu.be/tQDZXN_k5NU`.

**Keywords:** Mobile Manipulation, Legged Manipulator, Reinforcement Learning

## 1   Introduction

Legged manipulator robots offer significant potential in combining the ability of legged robots to navigate varied environments with the potential for rich interactions afforded by the manipulator. An essential skill for these robots is the autonomous opening and traversal of doors, significantly expanding their reach in human-centered environments. While door opening seems a quotidian task, it poses a challenge for control, especially for a high degree-of-freedom system such as a legged manipulator. Moreover, doors can vary in ways that are not immediately observable, such as in their spring stiffness and whether they are push or pull. The task becomes more difficult when the robot must also pass through the door as it must make additional decisions such as when to disengage from the handle or if it needs to hold the door open against the door's self-closing mechanism.

8th Conference on Robot Learning (CoRL 2024), Munich, Germany.

Existing methods for robot door opening have largely only considered being robust to properties that vary continuously such as the mass, spring resistance, or friction [1, 2, 3]. However, for determining the crucial property of the door opening direction, existing methods have relied on some hardcoded prior by either having a user provide the opening direction [4] or using pre-programmed routines [5]. These approaches limit the autonomy and adaptability of the robot in unknown environments. Ideally, the controller is "plug-and-play," meaning it does not rely on any priors and opens the door given only measurements of the door location by inferring the opening direction during the task, similar to how a person would open a door they have not encountered before. In moving towards this ideal, this paper proposes a learning-based legged manipulator control policy trained using the popular teacher-student approach [6] that can open and traverse through doors of varying properties via estimating these properties during deployment.

Reinforcement Learning (RL) with domain randomization is used to train the teacher policy as approaches such as imitation learning or model-based control are difficult to scale to doors of varying properties. The student policy is then trained to imitate the teacher and estimate the door properties using only the information available during deployment. To the best of our knowledge, our approach is the first monolithic control policy that can handle both push and pull doors without being given this information a priori or relying on any hardcoded routines to determine the door opening direction.

Our main contributions are summarized as:

- A control policy that can open and traverse through doors of varying characteristics (e.g. opening direction, dimensions, dynamical properties), given only the robot proprioception and the locations of the handle and doorway.
- Real-world hardware experiments validating the policy's efficacy on different doors.
- Analysis of the policy's ability to estimate the door properties from the task interaction.

## 2 Related Work

### 2.1 Model-Based Door Opening

Several works studied door opening for a manipulator on a wheeled base by tracking predefined reference velocities and forces [7, 8, 9, 10]. For legged manipulators, Model Predictive Control (MPC) has been applied for door opening [11, 2, 12]. MPC controllers rely on tracking a reference trajectory, which necessitates using a planner and requires knowing the properties of the door such as its opening direction and precise dimensions. As such, the practical applicability of such controllers to unseen doors is limited. The commercially available Spot robot with the arm manipulator includes a door opening controller. While its details are not public, it is likely a model-based controller. According to available documentation [4], Spot's controller must be provided information a priori such as the door's opening and swing directions.

### 2.2 Learning-Based Door Opening Control

RL is a natural approach for door opening as specifying the desired behavior through rewards for opening doors is intuitive. Several works used door opening as a benchmark task for RL algorithms [13, 14, 15]. Urakami et al. [1] developed the DoorGym simulation environment for RL door opening with a specific focus on robustness and generalization via domain randomization and demonstrated their trained policies in hardware experiments on a fixed-based manipulator. Schwarke et al. [3] trained and deployed an RL door opening policy on a wheeled-legged robot. However, these works produce control policies that can only handle a single opening direction as they treat push and pull doors as separate RL tasks. The works above have focused on opening the door and ignoring the subsequent task of passing through. Ito et al. [16] demonstrated a learning-based approach for opening and passing through doors with a wheeled base mobile manipulator. Their approach combines separate control modules for handling opening and passing through. However, the robot learns its motions from human demonstrations via teleoperation which is difficult to scale to different doors. Moreover, the method requires two separate modules for handling push and pull doors.

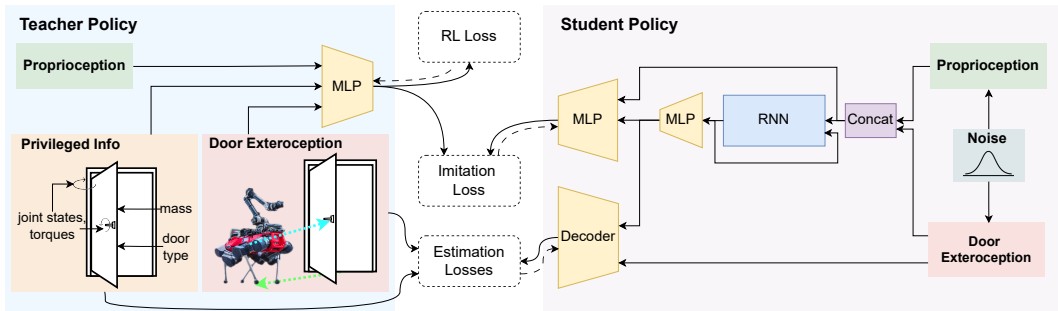

Figure 2: Overview of the training method. Dashed lines indicate the flow of gradients from the losses. The teacher policy is first trained using RL operating on a privileged set of observations. The student policy operates only on the observations available during deployment and is trained to imitate the teacher's behavior while also estimating the privileged information.

More recently, Kang et al. [5] combined RL and position-force controllers to open and pass through both push and pull doors with a wheeled base mobile manipulator. However, the RL controller is only used for push doors as the authors could not train a single RL policy to handle both opening directions. The method also relies on a hardcoded push-pull procedure after turning the handle to determine the door's opening direction.

## 2.3 Teacher-Student Distillation

Recent successes in applying teacher-student training to legged robot locomotion [17, 18] have demonstrated remarkable robustness and the student policy's ability to estimate properties of the environment. For example, in the locomotion task, the student policy can infer the height of the local terrain, even in the presence of corrupted exteroceptive sensor measurements by leveraging the knowledge of the feet's position. For door opening, the information necessary to complete the task can only be acquired through direct interaction with the door (e.g. push or pull). Given this, the teacher-student approach could be particularly beneficial for this task.

## 3 Method

An overview of our method is shown in Fig. 2. Both teacher and student policies are trained in a simulation environment in Isaac Gym [19] as shown in Fig. 3. The robot modeled in simulation and used in real-world experiments is ANYmal with an arm manipulator [20].

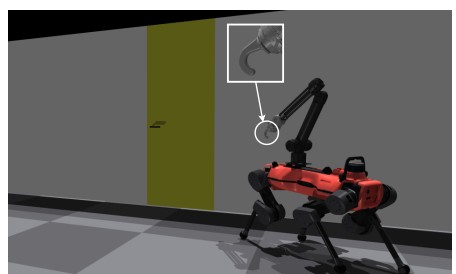

Figure 3: Overview of the training environment in simulation. A hook end-effector is used for grasping the door handle.

### 3.0.1 Door Model

We focus on hinged doors that consist of a single door panel with a handle for unlocking the door. The door has two degrees of freedom corresponding to the hinge and handle angles $\theta$ and $\phi$ respectively. To manipulate the handle, the robot is equipped with a hook end-effector. The handle must be turned to unlatch and open the door. Pretensioned spring resistances that act against opening the door and turning the handle are modeled as constant torques. The effect of a door closer system installed to regulate the speed of door closure was modeled as a damping torque on the door hinge that scales linearly with the hinge velocity. We included additional damping that scales quadratically with the hinge velocity to account for air resistance. The training environment contains four door types from the set {pull, push} × {right, left}. Where right and left denote whether the hinge is on the right or left side of the door panel.

### 3.0.2 Policy Actions

The policy directly controls the robot's arms and commands a lower-level locomotion policy that governs the legs. We use the learned locomotion policy from Ma et al. [21] which takes as input the planar linear velocities $v_x$ and $v_y$, angular velocity about $z$ in the robot base frame, and the applied wrench on the robot base from the arm and its motion. Instead of estimating the applied wrench, we found it sufficient and simpler to set it to a fixed mean value. To prevent the policy from moving the base too fast, we clip the linear and angular velocity commands to the locomotion controller to maximum magnitudes of 0.5 m/s and 1 rads/s respectively. Conversely, commands below 0.1 m/s and 0.1 rads/s are set to zero. The policy controls the arm by setting joint angle targets for the arm's joint-level PD controllers. The commanded target angle for joint $i$ is computed as

$$\text{clip}(sa_i + \tilde{q}_i, q_i - \sigma\bar{\tau}_i/K_p, q_i + \sigma\bar{\tau}_i/K_p), \tag{1}$$

where $a_i$ is the action, $\tilde{q}_i$ is default position, and $q_i$ is the joint position for joint $i$. Finally, $s$ is an action scaling factor. For safety, the commanded target for joint $i$ is clipped based on a proxy for the actuator torque threshold computed from the joint's proportional gain $K_p$, torque limit $\bar{\tau}_i$, and a saturation parameter $\sigma$. We set $\sigma$ to 0.7 for all arm joints. This action clipping is applied during training and on the real robot.

### 3.0.3 Sim-to-Real Considerations

The following were implemented in the simulation training environment to facilitate policy transfer onto the real robot. Detailed domain randomization parameters are reported in the Appendix.

*Randomize Initial Robot Configuration*: The robot base starts an episode in a random location and yaw angle in front of the door with a random initial base velocity. The initial arm configuration is not randomized to avoid self-colliding configurations.

*Randomize Arm Joint PD Gains*: The PD gains for each arm joint are resampled for each episode.

*Randomize Door Dimensions*: We generate door models while randomizing the handle locations on the door panel, the doorway width, and the door panel thickness.

*Randomize Door Dynamics Properties*: Resistance torques and the damping coefficients at the handle and hinge, the door mass, and the maximum handle turning angle are resampled for each episode.

*Zero Handle Contact Friction*: Many door handles have slippery surfaces which are especially hard to turn with a hook end-effector. We address this by zeroing the friction coefficient of the handle and hook end-effector contact in simulation. With non-zero handle contact friction, the policy tended to learn more aggressive behaviours that relied on the end-effector momentum to turn the handle.

## 3.1 Teacher Training

The teacher is trained as an Actor-Critic using Proximal Policy Optimization [22]. We separate the task into two stages. The first is to approach and open the door and the second is to pass through the door. This is done as the rewards are simpler to define when considering each stage in isolation.

### 3.1.1 Observations

The teacher observation includes measurements that are available to the robot during deployment along with privileged information only accessible in simulation. No noise is introduced to the teacher observation. The measurements available during deployment are the robot's proprioception and exteroceptive measurements of the location of the doorway and handle relative to the robot. Proprioception includes the base orientation, base linear and angular velocities, arm joint positions and velocities, and the previous step actions. The privileged observations include the door joint (hinge and handle) positions and velocities, the mass of the door panel, the applied torques at the door hinge and handle due to spring stiffness and damping, the door type, and a task stage observation denoting if the policy is currently in the stage of opening or passing through the door.

### 3.1.2 Rewards

The opening and passing stage rewards are denoted by $r_o$ and $r_p$ respectively. $r_o$ is further decomposed into rewards for handle manipulation, $r_{hm}$, and a reward for opening the door to a target angle, $r_{od}$. $r_{hm}$ comprises reward terms for moving the end-effector to the handle, grasping the handle, and turning the handle. The policy does not need to interact with the handle once the door has been unlocked and opened enough. As such, we set $r_{hm}$ to its maximum value such that the policy ignores it once the door is opened enough. We considered the door opened enough at 30°.

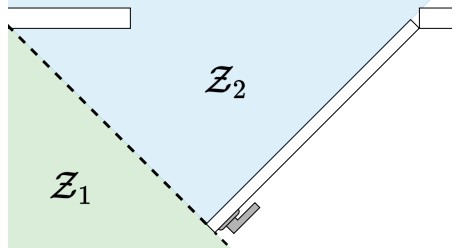

Figure 4: Additional rewards are used for pull doors to encourage the robot to move around the door panel. When the robot base is in $\mathcal{Z}_1$ it will receive reward $r_{\mathcal{Z}_1}$ and in $\mathcal{Z}_2$ it will receive reward $r_{\mathcal{Z}_1} + r_{\mathcal{Z}_2}$. These rewards are also given based on the end-effector location.

The opening stage transitions to the passing stage when the door has been opened more than 70°. After transitioning to the passing stage, the policy receives the maximum possible value of $r_o$ in addition to $r_p$. This is done so the policy can pursue $r_p$ without trading off $r_o$ as otherwise the policy can refuse to transition.

The passing stage reward $r_p$ is given by the dot product of the base velocity $\mathbf{v}_\mathcal{B}$ with the unit progress vector $\mathbf{p}$. Before the robot has passed through the doorway, $\mathbf{p}$ is the vector from the base to the doorway center. After the robot has passed through, $\mathbf{p}$ is the vector along the direction of the doorway. This dot product is normalized by the maximum commandable velocity of the locomotion controller $\|\mathbf{v}_\mathcal{B}\|_{\max}$. $r_p$ is clipped to a maximum of 1, otherwise, the policy can learn undesired behaviors such as throwing its arm to move the base faster than $\|\mathbf{v}_\mathcal{B}\|_{\max}$.

For pull doors, $r_p$ alone is insufficient for the policy to learn to move around the open door panel. Therefore, once a pull door is opened enough in the opening stage, we reward it for moving its base and end-effector around the door panel as shown in Fig. 4. We also require the base and end-effector to be behind the door panel to transition to the passing stage for pull doors.

Shaping rewards $r_s$ are used to regularize the behavior such that the policy respects the hardware limitations of the robot and is safe to deploy. $r_s$ is applied during both the open and passing stages and comprises terms such as avoiding collisions, minimizing arm motions, penalizing unwanted base motions, penalizing out-of-limit commands, and penalizing singular arm configurations.

Detailed definitions and scales of all reward terms are given in the Appendix.

## 3.2 Student Training

The student policy is trained to imitate the teacher policy's actions given only the proprioceptive and exteroceptive door observations that are available during deployment. Gaussian noise is added to all of the student's observations. The student policy also receives supervision through estimating the privileged information of the door and handle locations relative to the robot, the door joint states, the door mass, the door hinge and handle torques, and the door type. The Smooth L1 Loss is used for losses except for the door type where the Cross Entropy Loss is used.

The student policy is based on a recurrent neural network (RNN) with an architecture that follows the student policy for legged locomotion from Miki et al. [18] with some differences. Specifically, the attention-gate decoder in the original architecture is replaced with a linear layer as the decoder. This was done as unlike locomotion where it is possible to do blind, we assume that the exteroceptive measurements during door opening can be noisy but not degraded to the point of being useless.

We do not include all available measurements in the student policy observation similar to Tan et al. [23]. Specifically, the previous actions and arm joint velocities are omitted. Training the student policy with much higher noise on the arm joint velocity resulted in better transfer of the policy to the real robot. Given this, we completely removed the arm joint velocity from the observation.

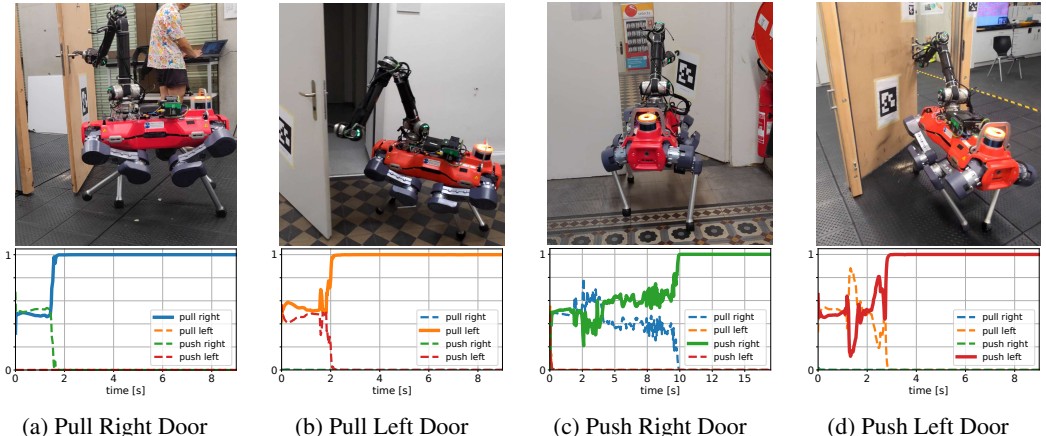

| (a) Pull Right Door | (b) Pull Left Door | (c) Push Right Door | (d) Push Left Door |

Figure 5: Real-world experiments of the policy deployed on the real robot, traversing through doors of varying swing (left/right) and opening (push/pull) directions. The policy's estimated probability for each door type over time is plotted below the corresponding experiment. The true door type is plotted as a solid line while others are plotted as dashed lines.

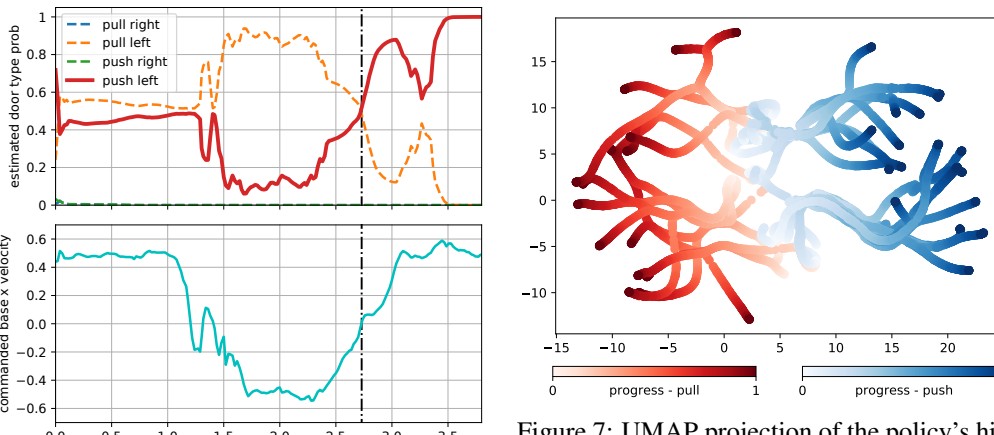

Figure 6: Correlation of the policy's estimated door type and its actions. At 2.7 seconds, the policy shifts its belief from pull to push and accordingly commands the base forwards to push.

Figure 7: UMAP projection of the policy's hidden state over trajectories collected in simulation with pull right and push right doors. All hidden state trajectories start at the central cluster and progress in separate directions depending on whether the door is push or pull.

## 4 Results

The student policy is deployed directly onto the robot without further fine-tuning. The policy is both trained and deployed at 50Hz. We tasked the robot with opening and traversing through doors varying in their opening and swing directions, doorway and handle dimensions, door panel inertia, handle dynamics, and the presence of a door self-closing mechanism.

Proprioceptive observations are provided by sensors onboard the robot, such as encoders measuring the joint states. The robot also runs onboard lidar odometry [24] to estimate its base pose. For exteroception, the policy only needs to know the handle and doorway locations relative to the robot base. We provide these door measurements using either motion capture or AprilTags [25] as an external tracking system. The motion capture system provides accurate measurements but requires several fixed tracking cameras where as the AprilTags can be easily deployed on any door by tracking a tag on the door panel with an external camera. As the door measurements are relative to the robot, they could also be obtained using onboard sensing, but we leave this as future work.

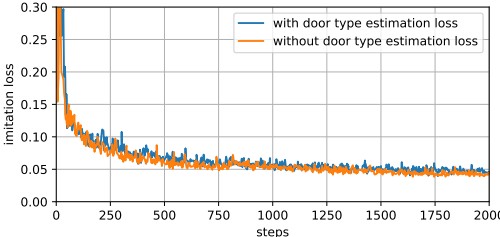

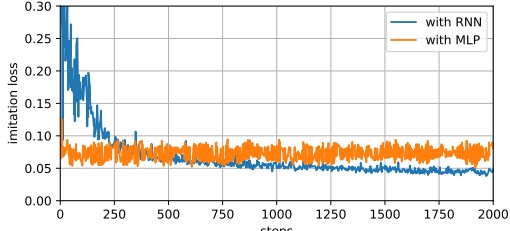

Figure 8: Comparing imitation loss when training the student policy with and without the door type estimation loss.

Figure 9: Comparing imitation loss when training student policy with and without a recurrent module.

## 4.1 A Single Control Policy for All Door Types

We evaluate whether the policy can recognize the door type (opening and swing directions) during the task and open them without being given this information beforehand. Indeed our policy could successfully deduce the correct type and pass through for all door types as shown in Fig. 5. When the policy was uncertain of the opening direction, it shifted its estimate between push and pull. This shift of the policy's belief was correlated to its commanded actions as shown in Fig. 6, where the policy learned to push and pull until it could move the door panel. We also study how the policy's belief of the door type is represented in its RNN hidden state by visualizing the hidden states trajectories using a UMAP [26] projection in Fig. 7. The figure shows that the policy learned to represent the opening direction (push/pull) in separate regions of the hidden state space. Moreover, the regions for the opening directions appear linearly separable in the projection.

## 4.2 Repeatability of the Control Policy

Repeated trials of the policy were conducted to evaluate its repeatability. A spring-loaded door was placed within the motion capture space and 20 continuous trials were performed for both the pull and push sides. The policy successfully traversed the door in 20/20 and 18/20 trials for the pull and push sides respectively, with an overall success rate of 95.0%. In all trials, the policy was successfully able to open the door. The two failed trials on the push side were caused by the robot getting stuck on the side of the doorway which was protruding due to a lack of walls around the door which differs from the simulation model. A video of the trials is provided in the supplementary material

## 4.3 Ablations in Simulation

We ablated the student policy training to study if supervision from the estimation task helps in learning the control task and if the recurrence of the student policy is necessary.

*Door Type Estimation Loss*: The student policy is trained without supervision from estimating the door type and we find that the imitation loss alone is sufficient for learning to distinguish between both push and pull doors. Moreover, removing the estimation loss has little effect on the training dynamics of the imitation loss as seen in Fig. 8, but its presence neither helps nor hurts the imitation learning. Given this, the estimation loss and decoder module are still included in our final policy architecture as they are useful for helping to understand the internal belief state of the policy.

*Student Policy Recurrence*: We replaced the RNN module of the student policy with an MLP and trained the policy with the imitation loss to study the effect of removing recurrence. Fig. 9 shows that the student policy fails to imitate the teacher policy without recurrence. Specifically, the MLP student policy cannot imitate the teacher given the noisy student observations. As the MLP operates on a single-step observation, any noise on the observation makes it difficult for the policy to disambiguate different states of the system and to take the appropriate action.

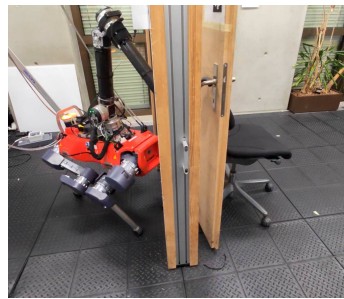 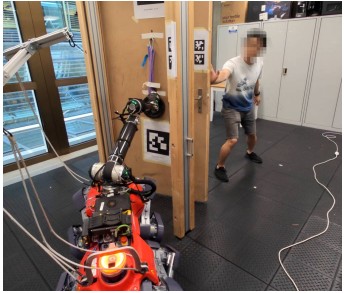 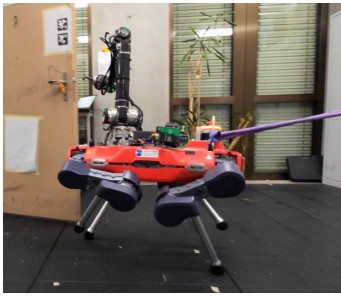

| (a) Chair blocking the door. | (b) Human pushing on the door. | (c) Rope pulling on the robot. |

Figure 10: Robustness of the policy to external disturbances unmodelled in the training environment.

## 5 Discussion

### 5.1 Emergent Robustness to Unmodeled Disturbances

The student policy showed robust recovery from disturbances not explicitly modeled during training in simulation. For example, the policy could handle pushing or pulling on the door or robot as seen in Fig. 10, even though random pushes/pulls on the robot were not used during training. It can be expected that the teacher policy would be robust to such disturbances as it acts on the current observation without regard to the history. As the student does not perfectly imitate the teacher during training, mistakes in the student's actions during training provide opportunities to learn recovery behaviours from the teacher. For example, if the student's action missed the handle grasp, the subsequent teacher's actions to imitate would be to retry grasping the handle.

### 5.2 Failure Cases

A common failure case of the policy was not turning the handle enough to unlock the door. When this failure occurs, the policy becomes stuck moving the robot back and forth trying to push and pull on the door as it shifts its belief of the door type. The policy does not have a direct observation which indicates that the handle has been turned. As such, it can learn to incorrectly associate the robot's handle turning motion with the handle turning. Using visual or force sensing, which can directly observe the turning of the handle, could alleviate this issue.

The performance of the policy was notably degraded when using AprilTags to track the door due to characteristics of the AprilTag measurement noise being unmodeled in simulation when training the policy. We note a failure case caused by the AprilTag latency where the policy lets go of the handle and attempts to grasp it again as it receives delayed measurements of the handle location.

Lastly, unmodeled door geometry in simulation causes failure cases such as the hook end-effector catching on the door panel or the robot base getting stuck on the protruding sides of a doorway as shown in the videos provided in the supplementary material.

## 6 Conclusion

This paper presented a learned control policy for a legged manipulator to traverse through doors. The policy was trained in simulation using a teacher-student approach such that it learned both robust behaviors and the ability to estimate properties of the door through interaction during the task. The latter allowed the policy to handle doors of varying properties, such as the opening direction, without being given this information a priori. The policy was deployed on the ANYmal robot with an arm and achieved a 95.0% success rate in trials on the push and pull sides of a spring-loaded door. Additional experiments demonstrated the policy traversing through doors of all opening and swing directions, various dimensions, and various hinge and handle dynamics, as well as robustness to and recovery from external disturbances. Future works include using onboard sensing for obtaining the door measurements, adding force sensing, and handling a larger set of door handles such as knobs.

## Acknowledgments

We would like to thank Eris Sako, Jan Preisig, Jia-Ruei Chiu, Kaixian Qu, and Turcan Tuna for helping with experiments and Andrei Cramariuc for feedback on the paper. This work was supported by the Max Planck ETH Center for Learning Systems, Intel, and NCCR digital fabrication. Moreover, this work was conducted as part of ANYmal Research, a community to advance legged robotics.

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

# Appendix

## A    Reward Definitions

### A.1    Opening Rewards

The opening reward $r_o$ is composed of the reward for opening the door to the target angle $r_{od}$ along with rewards for manipulating the door handle denoted by $r_{hm}$. For pull doors, we additionally include a reward for encouraging the robot to move its base and end-effector around the door panel denoted by $r_{adp}$. These rewards together compose the opening reward

$$r_o = 3r_{od} + \begin{cases} r_{hm} & \theta < 30^o \\ \bar{r}_{hm} + 0.5r_{adp} & \text{otherwise} \end{cases}$$

When the door has been opened enough, set as $\theta > 30^o$, $r_{hm}$ is set to its maximum value $\bar{r}_{hm}$ as it is no longer necessary for the policy to interact with the handle to open the door further. $r_{adp}$ is only applied once the door has been opened enough.

The handle manipulation reward is composed of the following components

$$r_{hm} = r_{ehd} + r_{th} + r_{eho} + 0.5r_{hg} + r_{plg}$$

All individual reward terms in $r_o$ are defined as follows.

- $r_{ehd}$ (end-effector to handle): Minimizes the distance between the end-effector point $\mathbf{e}$ and the handle point $\mathbf{h}$:
$$r_{ehd} = \exp(-\|\mathbf{e} - \mathbf{h}\|_2)$$

- $r_{th}$ (turn handle): Rewards increasing the handle turning angle $\phi$:
$$r_{th} = \phi/\phi_{\max}$$
where $\phi_{\max}$ is the maximum the handle can be turned.

- $r_{eho}$ (end-effector grasp orientation): Rewards the end-effector for tracking a desired orientation for grasping the handle.
$$r_{eho} = 1 - \frac{|e_o|}{\pi}$$
where $e_o$ angular error between the end-effector orientation and the desired end-effector orientation.

- $r_{hg}$ (handle in end-effector grasp): Give a binary reward when the handle point $\mathbf{h}$ is within the grasp zone $\mathcal{G}$ of the end-effector.
$$r_{hg} = \begin{cases} \mathbf{1}_{\mathcal{G}}(\mathbf{h}) & \|\mathbf{e} - \mathbf{h}\|_2 \leq 1 \\ 0 & \text{otherwise} \end{cases}$$

  $\mathbf{1}_{\mathcal{A}}(x)$ is the indicator function of value 1 if $x \in \mathcal{A}$ and 0 otherwise. For the hook-end effector used in this work, we defined the grasp zone $\mathcal{G}$ as the region along the opening of the hook. This reward is only active when the end-effector point $\mathbf{e}$ is close enough to the handle (within 1 m).

- $r_{plg}$ (penalize lost grasp): Give a binary penalty when if the handle point $\mathbf{h}$ is in the grasp zone at step $t - 1$ and leaves the grasp zone at $t$.
$$r_{plg} = \begin{cases} -\mathbf{1}_{\mathcal{G}}(\mathbf{h}_{t-1})(1 - \mathbf{1}_{\mathcal{G}}(\mathbf{h}_t)) & \|\mathbf{e} - \mathbf{h}\|_2 \leq 1 \\ 0 & \text{otherwise} \end{cases}$$
Similar to $r_{hg}$, $r_{plg}$ is only active when the end-effector is close enough to the handle.

- $r_{od}$ (open door to target angle): Rewards opening the door to the target opening angle $\bar{\theta}$
$$r_{od} = 1 - \frac{|\theta - \hat{\theta}|}{\hat{\theta}}$$
where $\theta$ is the door hinge joint angle. This reward can also be used to train an opening only policy that opens the door to $\hat{\theta}$. For training the opening and passing through policy $\hat{\theta}$ is set to $75°$.

- $r_{adp}$ (move around the door panel): This reward is only applied for pull doors.

  Given zones $\mathcal{Z}_1$ and $\mathcal{Z}_2$ defined relative to the door panel as shown in Fig 4, $r_{adp}$ is computed based on the locations of the base $\mathbf{b}$ and the end-effector $\mathbf{e}$ as

$$r_{adp} = \begin{cases} 1 & \mathbf{b} \in \mathcal{Z}_1 \\ 2 & \mathbf{b} \in \mathcal{Z}_2 \\ 0 & \text{otherwise} \end{cases} + \begin{cases} 1 & \mathbf{e} \in \mathcal{Z}_1 \\ 2 & \mathbf{e} \in \mathcal{Z}_2 \\ 0 & \text{otherwise} \end{cases}$$

## A.2  Passing Rewards

- $r_p$ (passing progress): Rewards the base velocity $\mathbf{v}_\mathcal{B}$ for moving along the unit progress vector $\mathbf{p}$

$$r_p = \max\left(1, \frac{\mathbf{p} \cdot \mathbf{v}_\mathcal{B}}{\|\mathbf{v}_\mathcal{B}\|_{\max}}\right)$$

  where $\|\mathbf{v}_\mathcal{B}\|_{\max}$ is the the max allowable commanded velocity of the locomotion controller.

## A.3  Shaping Rewards

The shaping reward $r_s$ is defined as

$$r_s = 0.3r_{ma} + 0.5r_{pbt} + r_{psa} + 0.1r_{pcl} + 2r_{pc}$$

Individual terms of $r_s$ are defined as:

- $r_{ma}$ (minimize arm motion): Rewards minimizing the arm joint velocities and accelerations

$$r_{ma} = \sum_{i=1}^{6} \exp(0.01\dot{q}_i^2) + \exp(0.000001\ddot{q}_i^2)$$

  where $\dot{q}_i$ and $\ddot{q}_i$ are the joint velocity and acceleration for the $i^{th}$ arm joint respectively.

- $r_{pbt}$ (penalize base tilt): Penalizes large tilt of the robot base. The base tilt angle $\psi$ can be computed from the projected gravity vector expressed in the robot base frame $\mathbf{g}_\mathcal{B}$ and expressed in the world frame $\mathbf{g}_\mathcal{W}$ as follows

$$\psi = \arccos\left(\frac{\mathbf{g}_\mathcal{W} \cdot \mathbf{g}_\mathcal{B}}{\|\mathbf{g}_\mathcal{W}\|\|\mathbf{g}_\mathcal{B}\|}\right)$$

  Then $r_{pbt} = -1$ if $\psi > \bar{\psi}$, where $\bar{\psi}$ is a tilt threshold, and 0 otherwise. We set $\bar{\psi}$ as $8°$.

- $r_{psa}$ (penalize stretched arm): Penalize the arm from reaching out too far to prevent singular arm configurations.

$$r_{psa} = -\text{clip}\left(\frac{\|\mathbf{e} - \mathbf{s}\| - (0.7 - 0.1)}{0.1}, 0, 1\right)$$

  where $\mathbf{e}$ and $\mathbf{s}$ are the locations of the end-effector and shoulder joint.

- $r_{pcl}$ (penalize command out of limits): As the arm PD target and locomotion commands are clipped within certain bounds we penalize the policy for commands that exceed these bounds.

$$r_{pcl} = -\sum_{i=1}^{9} \text{clip}\left(\frac{|a_i| - \bar{a}_i}{\sigma_i}, 0, 1\right)$$

  where $a_i$, $\bar{a}_i$, and $\sigma_i$ corresponding to the action, action limit, and penalty ramp up speed for the $i^{th}$ component of the policy's output action. The action limits are discussed in Sec. 3.0.2.

- $r_{pc}$ (penalize collisions): Penalizes robot collisions.

$$r_{pc} = -\sum_{c \in \mathcal{C}} \left(\begin{cases} 1 & \|\lambda_c\| > 0 \\ 0 & \text{otherwise} \end{cases}\right)$$

  where $\lambda_{(\cdot)}$ is the contact force on robot link $(\cdot)$ and $\mathcal{C}$ is the set of robot links where collisions are penalized including the base, thighs, and arm.

# B Domain Randomization Parameters

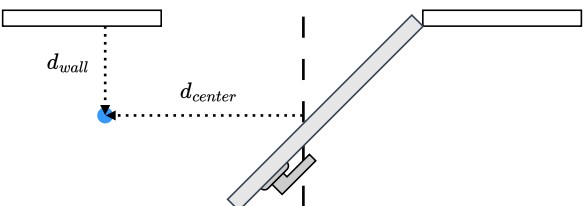

Figure 11: The initial robot location relative to the door is determined at the beginning of each episode by sampling $d_{wall}$ and $d_{center}$.

The following randomizations are resampled for each new episode:

- Initial Base Location: Set relative to the doorway by the distances $d_{wall}$ and $d_{center}$ as shown in Fig. 11. $d_{wall}$ and $d_{center}$ are sampled uniformly from $[1, 2]$ m and $[-2, 2]$ m respectively.
- Initial Base Yaw: We define a yaw of $0^o$ as the robot facing forwards along the direction of the doorway. The initial yaw is sampled uniformly from $[-180, 180]^o$.
- Initial Base Velocity: The initial base velocity components $v_x$ and $v_y$ are sampled uniformly from $[-0.5, 0.5]$ m/s.
- Door Panel Mass: Sampled uniformly from $[15, 75]$ kg.
- Door Hinge Resistance Torque: Sampled uniformly from $[0, 30]$ Nm, set to 0 with probability $0.2$.
- Door Handle Resistance Torque: Sampled uniformly from $[0, 3]$ Nm, set to 0 with probability $0.2$.
- Door Hinge Damping Torques: The hinge damping torque comprises of the air resistance given by $K_d^{ar}\dot{\theta}^2$ and the door closer mechanism damping given by $K_d^{dc}\dot{\theta}$. We sample $K_d^{ar}$ uniformly from $[0, 4]$ Nms$^2$ For most doors, the door closer's damping is tuned to prevent the door from closing too quickly. To model this, we set $K_d^{dc}$ to be some multiple $\alpha$ of the hinge resistance torque, where $\alpha$ is sampled uniformly from $[1.5, 3]$ s. The hinge damping torque is set to 0 with probability $0.4$.
- Maximum Handle Turning Angle: Sampled uniformly from $[15, 90]^o$.
- Arm Joint Proportional Gain: Sampled uniformly from $[40, 60]$.
- Arm Joint Damping Gain: Sampled uniformly from $[3, 6]$.

We generated door models with different dimensions that are loaded into the simulation during initialization. The randomized door dimensions are shown in Fig. 12.

- $d_W$: Sampled uniformly from $[0.8, 1.0]$ m.
- $d_T$: Sampled uniformly from $[0.02, 0.06]$ m.
- $h_L$: Sampled uniformly from $[0.08, 0.12]$ m.
- $h_H$: Sampled uniformly from $[0.7, 1.3]$ m.
- $h_O$: Sampled uniformly from $[0.03, 0.12]$ m.

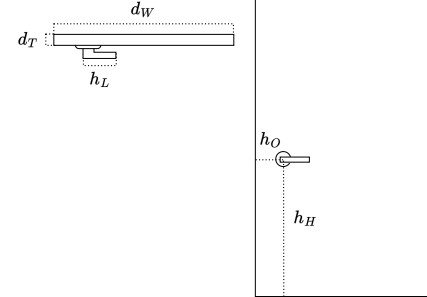

Figure 12: Randomized dimensions of the door and handle.

# C    Additional Experiments

## C.1    Is Teacher-Student Training Necessary?

To evaluate the necessity of the teacher-student framework for this task we trained two modified teacher policies to see if a policy trained with RL directly on the observations available on the robot can learn the task. The first is given only the student observations without access to the privileged observations. The second is the same as the first except that noise is added to the observations at the same amount used for training the student. We evaluated the teacher policies in simulation on 4000 environments (a different randomized door in each) for 10 episodes, each lasting 10 seconds, with all domain randomizations active. The success rates are reported in Fig.13.

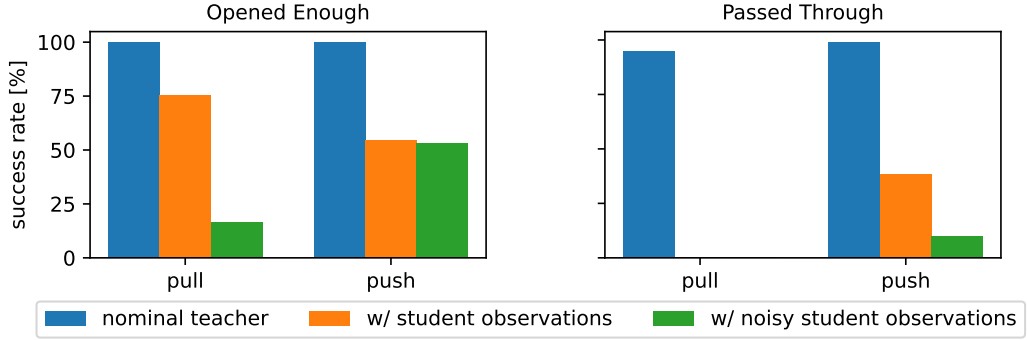

Figure 13: Comparing the nominal teacher policy against variants with only the student observations. The performance is reported as success rates on whether the policy could open the door enough and if it could pass through the door. We consider the policy to have opened the door enough if the door was opened by at least $30°$ within an episode. The policy is considered to have passed through the door if the robot reaches a position 0.5 meters behind the door wall.

We find that without the privileged observations, the teacher policy could not learn to pass-through for both opening directions, but is surprisingly able to learn an opening behavior that can sometimes open both push and pull doors, albeit at a significantly worse success rate. Examining this behavior qualitatively suggests that without the door type privileged information, the policy learns to associate specific robot configurations with either push or pull, learning to move between these configurations until the door opens. Adding noise to the observations further reduces the success rate for this type of behavior as disambiguating these specific robot configurations becomes more difficult.

## C.2   Evaluating the Student Policy's Capabilities

The capability of the student policy is evaluated in simulation by tasking it with traversing through doors of increasing hinge resistance torques. Increasing the hinge resistance makes the doors more difficult to open as the robot must apply more force before the door moves, and more difficult to pass through as the door closes faster after opening. The student policy is tested on whether it could open the door and if it could successfully pass through in episodes lasting 10 seconds with all domain randomizations active. We report these results in Fig.14.

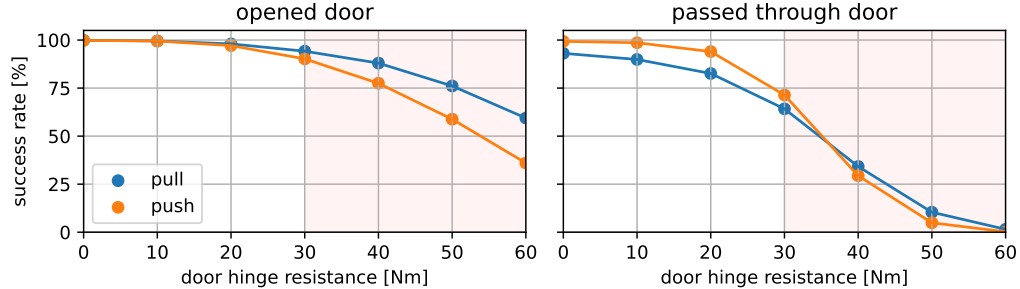

Figure 14: Evaluating the capability of the student policy on doors of increasing hinge resistance torques. The performance is reported as success rates on whether the policy could open the door enough and if it could pass through the door. We consider the policy to have opened the door enough if the door was opened by at least $30°$ within an episode. The policy is considered to have passed through the door if the robot reaches a position 0.5 meters behind the door wall. During training, the policy did not experience hinge resistance torques greater than 30 Nm (highlighted in red).

The teacher and student policies were trained with a hinge resistance randomization range of 0 Nm to 30 Nm. As expected, the success rates for both opening and passing through the door decreases for increasing hinge resistance. For the doors with greater hinge resistances, the policy could still sometimes open them but passing through is more difficult with the success rate dropping to near zero for hinge resistances greater than 50 Nm. For push doors, the higher resistance makes the door panel more difficult for the robot to hold open and the robot can also get trapped in the door way by the door panel. For pull doors, the higher resistance causes the door to close faster, making it more difficult for the policy to move the robot around the door panel in time to pass through.

