# OpenReview forum: "Learning to Open and Traverse Doors with a Legged Manipulator"
_robot-learning.org/CoRL/2024/Conference — CoRL 2024_

### Official Review · Reviewer_AGnv · 2024-07-16
**A practical learning-based method for door opening in real-world environments**

**Originality:** 4
**Technical Quality:** 5
**Clarity Of Presentation:** 3
**Potential Impact:** 3
**Recommendation:** 3
**Confidence:** 3

**Review:**

Strengths:
1. This paper is well-motivated, as door opening and traversal is a crucial skill for mobile manipulation.
2. The paper is clearly written and includes sufficient technical details, allowing for reproducibility.
3. The experiments are conducted entirely on real robot hardware, demonstrating the approach's effectiveness in real-world scenarios. The evaluation also considers domain randomization and human disturbance, showcasing the trained policy's robustness.

Weaknesses:
1. The trained policy assumes knowledge of the handle and doorway locations relative to the robot, which could be a significant limitation for deployment in uncontrolled environments.
2. The total number of experiments is limited, with roughly half involving pulling doors and the other half pushing. While there are demonstrations of domain randomization and human disturbance, there is no formal statistical analysis of the results.
3. (Minor) The reviewer observed a hook as the robot's end-effector, which may limit its capability to perform other manipulation tasks.

**Quality Of The Limitations Section:**

2

**Questions For Rebuttal:**

1. The reviewer is curious about the circumstances under which the learned policy works best. For example, does it perform better when pulling or pushing doors, and what is the maximum door weight the robot can handle? Could these conclusions be supported by statistical analysis, such as conducting the same set of experiments multiple times or even in simulation?
2. Can the author further justify the need for the teacher-student policy training framework through an ablation study?
3. What is the reason behind the hardware design choice of a hook instead of a two-finger gripper?

**Robotics Focus:**

4

**Summary Of Paper:**

This paper aims to address the challenging and long-standing problem of mobile manipulation, specifically door opening. The proposed approach employs teacher-student distillation policy training, enabling the robot to learn an open-and-traverse policy using reinforcement learning (RL) without human demonstrations. The learned policy is deployed and evaluated on real robot hardware (i.e., ANYmal), demonstrating its effectiveness across various door-opening tasks. It shows generalized capabilities for different opening directions (push and pull), door dimensions, masses, and other dynamic properties.

**Summary Of Recommendation:**

The paper is highly engineering-driven. While it does not introduce significant novelty from a research perspective, it demonstrates significant improvements in the robot's hardware capabilities.

---

### Official Review · Reviewer_wfP1 · 2024-07-21
**Well written paper solving complex robot door opening task in real world**

**Originality:** 4
**Technical Quality:** 4
**Clarity Of Presentation:** 5
**Potential Impact:** 3
**Recommendation:** 4
**Confidence:** 5

**Review:**

The paper is well written, easy to understand and the authors do an excellent job of describing the details of the experiment. The problem tackled by the authors, which is to train a single policy which can open variety of doors as well as pass through it, is very relevant to deploying robots in real world environments and not explored much in earlier literature. The compelling results make it a strong contribution to the field.  The limitations and future work discussions are also satisfactory.

Some of the strengths of the paper are
1) The results are well reported with nice visualizations. For example, it was helpful to see the door estimation probabilities.
2)  Strong real world results with very high success rate.

In my opinion, some weaknesses
1) The proposed method relies on careful reward design with many terms for shaping the robot behavior as well as careful selection of domain randomization parameters to enable sim-to-real. I wonder how scalable the approach is so that it generalizes to perhaps different robot / end-effector/ door handle type.
2) While the door type estimation output was presented in the paper, the performance of the estimator on the other door properties like handle location and door joint angles could be helpful to the reader. Even if it is in simulation.
3) Supplementary videos are detailed and nicely done.

**Quality Of The Limitations Section:**

3

**Questions For Rebuttal:**

1) At what frequency was the policy executed? Was it the same in simulation and real world. In general, more details on efforts to transfer the policy might be helpful. Was domain randomization sufficient?
2) From what I understand, the 95% success rate reported was using the motion capture system for the door exteroception? Please correct me if Im wrong. The authors also mention that the performance degrades when using AprilTags, how much worse was the performance?

**Robotics Focus:**

4

**Summary Of Paper:**

The authors present a Teacher-Student reinforcement learning framework to learn a single policy capable of opening different types of doors (left/right opening, push/pull). A key component that sets the paper apart from existing literature is that the robot not only opens the door but the same policy also guides the robot past the open door. The results are also evaluated on real world with impressive results. This is achieved through careful reward design in simulation and domain randomization which transfers to real robot well.

**Summary Of Recommendation:**

Accept

---

### Official Review · Reviewer_26YG · 2024-07-25
**The paper pushes the use of student-teacher architecture to a different task than locomotion and demonstrates good results.**

**Originality:** 4
**Technical Quality:** 4
**Clarity Of Presentation:** 5
**Potential Impact:** 4
**Recommendation:** 4
**Confidence:** 5

**Review:**

## Strengths
1. The paper is well written and the problem is relevant to the robot learning community as this work can be applied to other robots, for ex, humanoids, mobile manipulators.
2. The architecture for the student-teacher framework is novel -- the addition of the decoder facilitates interpretability as illustrated in the paper.
3. The reward structure is novel and shows how you can combine two sequential tasks into one RL problem.
4. The supplementary material also goes in depth into the implementation details such that these are sufficient to reproduce. The videos also show adversarial cases where the legged robot overcomes a chair blocking the opening of the door.
5. The push and pull scenarios are quite unique and useful to show the roll the proprioceptive feedback plays.

## Weaknesses
1. Design of the hook and the downward pull door handle may be restrictive for the robot on other kinds of doors or to do other tasks. Although the paper shows good performance, the reviewer see's this as a limitation where the method is limited to the design.
2. The door design maybe restrictive and goes against the "plug and play" claim. For doors designed differently, this may not work.

Nitpick:
- On line 180 "Th Smooth L1 loss"

**Quality Of The Limitations Section:**

2

**Questions For Rebuttal:**

The reviewer thinks the paper is clear and well written.

**Robotics Focus:**

4

**Summary Of Paper:**

The paper addresses the problem of opening doors with a legged manipulator. The proposed method uses the student teacher framework to train the policy and demonstrates the effectiveness of the method with zero shot sim2real transfer.  Different reward functions are used for the two tasks of opening the door and passing through it along with other rewards for locomotion. Ablation studies are provided to show how the different input parameters and the chosen architecture impacts the performance.

**Summary Of Recommendation:**

The paper is very clear and contributions are novel.

---

### Author Rebuttal · Authors · 2024-08-07

The rebuttal materials contain the revised paper which currently includes the following changes:
- Fixed typo pointed out by Reviewer 26YG
- Mentioned the policy frequency of 50Hz for both training and deployment based on feedback from Reviewer wfP1

We have also added to the rebuttal material a revised appendix which contains the following additional experiments suggested by Reviewer AGnv:
- Evaluating the necessity and benefits of using the teacher-student training framework for the door traversal task
- Benchmarking the capabilities of the policy in simulation

The rebuttal material also includes a video of the method used to train a policy with the two-finger EZGripper instead of the hook end-effector.

---

### Decision · Program_Chairs · 2024-09-04

**Decision:**

Accept

**Comment:**

This paper proposes a method for a legged robot with an arm that can traverse through a wide variety of doors. The reviewers were highly favorable. The method is still somewhat application specific (opening doors) but the results are an excellent contribution to the robot learning community.

Strengths:
- The policy can result in a legged robot opening doors and moving through the open door.
- There are substantial real world experiments.

Weaknesses:
- It requires careful reward design which might be hard to generalize to other scenarios.
- Further justification of student-teacher training could help for this particular pipeline. Also the technique itself is not novel as pointed out in the paper.